# The Influence of Proteins on Fate and Biological Role of Circulating DNA

**DOI:** 10.3390/ijms23137224

**Published:** 2022-06-29

**Authors:** Oleg Tutanov, Svetlana Tamkovich

**Affiliations:** V. Zelman Institute for Medicine and Psychology, Novosibirsk State University, 630090 Novosibirsk, Russia; ostutanov@gmail.com

**Keywords:** circulating DNA, cell-free tumor DNA, cell surface-bound DNA, DNA-binding proteins, extracellular vesicles, exosomes

## Abstract

Circulating DNA has already proven itself as a valuable tool in translational medicine. However, one of the overlooked areas of circulating DNA research is its association with different proteins, despite considerable evidence that this association might impact DNA’s fate in circulation and its biological role. In this review, we attempt to shed light on current ideas about circulating DNA origins and forms of circulation, known biological effects, and the clinical potential of circulating tumor deoxyribonucleoprotein complexes.

## 1. Introduction

The first demonstrations of DNA in the blood of healthy individuals date back to 1948 [1,2,3]. The colorful history of circulating DNA (cell-free DNA and cell surface bound DNA, which hereafter will be referred to as cirDNA) research and attempts of its use in the field of oncology went from being skeptically discarded to becoming a valuable tool in clinical oncology. Despite a big effort aimed at the elucidation of cirDNA origins, circulation forms, biological role, and clinical use, the nature of this phenomenon and its diagnostic potential remains poorly understood. Following multimarker initiatives, one of the approaches that can shake the existing views up is approaching cirDNA from a protein standpoint. Some approaches, such as the diagnostic method based on nucleosome binding patterns in tumor cirDNA, have already managed to do it, while others remain to be discovered [4].

## 2. Origins of cirDNA

Through cell death and active secretion, cirDNA is continuously shed into human blood in the form of membrane-containing structures (such as apoptotic bodies) or nucleoprotein complexes containing fragmented DNA molecules [5,6]. Despite intensive research, the contribution of each of these processes to the presence of cirDNA in the bloodstream is still unclear. Further degradation of cirDNA as part of nucleoprotein complexes occurs in the blood under the action of extracellular nucleases, while blood proteases play an equally important role, increasing the availability of DNA for hydrolysis by digesting proteins as part of nucleoprotein complexes.

### 2.1. Cell Death Origin of cirDNA (Apoptosis, Necrosis, and Autophagy)

At least 10^10^ cells are going through division stages daily [7]; during this process, some cells die, generating 1–10 *g* of DNA per day. One of the signs of apoptosis is the internucleosomal fragmentation of DNA (with fragment sizes corresponding to nucleosomes), followed by the formation of apoptotic bodies. Thus, the presence of cirDNA, ranging in size from 180 to 200 bp, in blood serum and plasma in normal conditions [8,9] and in the development of malignant neoplasms [10,11,12] might indicate its possible apoptotic origin. It has been demonstrated via in vitro and in vivo systems that Jurkat cells, in which apoptosis is induced, release significantly more cirDNA into the culture medium than non-apoptotic ones [13,14,15]. It has been shown that the release of DNA from apoptotic cells is a time-dependent process with the amount of DNA proportional to the degree of apoptosis, and that the pattern of the released DNA fragmentation is similar to the pattern of DNA fragmentation activated after the initiation of apoptosis by nucleases [13,14,15]. Apoptosis can be regulated by autophagic activity. Moreover, autophagy has been found to be activated in malignant cells [16]. Since autophagy is associated with the citrullination of histones to ensure the unwinding and subsequent expulsion of DNA, it can be assumed that this process also contributes to an increase in the concentration of cirDNA in the blood. 

With necrotic cell death, such pattern of fragmentation is not observed. Several studies have reported long cirDNA fragments (~10,000 bp) in the blood plasma of healthy donors and cancer patients, which might be indicative of necrotic cell death [9,10,12,17]. However, due to the absence of necrotic cells in a healthy state, necrosis cannot be the main source of cirDNA in healthy individuals [5]. 

It is known that phagocytic cells are heavily involved in the removal of both cellular debris [18] and apoptotic bodies [19]. In the study by Pisetsky’s group, Jurkat cells entering apoptosis were co-cultivated with macrophages, and this co-cultivation reduced the amount of DNA released in the culture medium. At the same time, macrophage co-cultivation with Jurkat cells with induced necrosis was characterized by high concentrations of cell-free DNA [20]. Thus, it can be assumed that some part of the cellular debris can avoid this fate and remain in the blood, increasing the concentration of cirDNA. 

### 2.2. Appearance of Blood cirDNA during Secretion by Normal and Tumor Cells 

Hypotheses about necrotic and apoptotic origins of cirDNA are contradicted by data indicating a decrease in the blood plasma cirDNA levels by 90% after the end of a course of radiotherapy [21]. It is well known that radiation induces necrosis (or apoptosis, depending on the radiation dose), hence such treatment should lead to an increase in cirDNA concentration.

There is a lot of published evidence on the active secretion of cirDNA by cells. However, questions about the mechanisms of such processes and possible pathways governing the secretion via either complexes with nucleosomes or DNA-containing extracellular vesicles (EVs) remain open. Abe and colleagues suggest a mechanism of cirDNA secretion indirectly with EVs [22]. It is an established fact that the total amount of tumor cirDNA (ctDNA) correlates with the stage of the disease, the number of metastases, and a decrease in overall survival [23,24,25]. Some studies described that the size of cirDNA is 160–170 bp, with long cirDNA fragments resulting from genomic DNA contamination [26]. However only long cirDNA fragments were detected in the cancer cell culture media (1000 to 10,380 bp), while in animal models using the same cell lines, both long (1000 to 10,380 bp) and short (130 to 240 bp) cirDNA fragments were detected [1,27]. Moreover, long cirDNA fragments (1 to 9 kbp) are found in breast cancer patients at the I–II stage [9] and in lung cancer patients with advanced stages of the disease [28]. Thus, cirDNA secreted by tumor cells can be degraded by nucleases in the peripheral blood since tumor cirDNA is detected in both long and short fragments [22,29]. Indeed, long cirDNA fragments were detected in the EV fraction (after 100,000× *g* centrifugation); the authors suggest that it is the association of cirDNA with EVs that protects it from degradation in the peripheral blood [22]. 

While the presence of DNA in the EV lumen is currently a very debatable question in the scientific community, there is increasing evidence of cirDNA and cirRNA present on the outer surface of small EVs (including exosomes—vesicles with a size of 30–150 nm, carrying tetraspanins such as CD9, CD63, and CD81 on their membrane) bound by nucleic acid binding proteins. In our previous work, we have shown the presence of DNA on the outer membrane of exosomes, and have identified several membrane DNA-binding and histone-binding proteins in small EV proteomes (AIFM1, IGHM, CHD5, KCNIP3, CHD5, and KDM6b, respectively) [30]. However, it was shown that the share of that cirDNA fraction did not exceed 0.025% in the total pool of cirDNA from the blood of healthy females and breast cancer patients, suggesting that this type of circulation barely contributes to cirDNA circulation in general. 

Another notable work, dedicated to the reassessment of exosome composition, echoes the aforementioned one [22,31]. The authors have shown that double-stranded DNA and histones H2a and H3 were absent in bona fide exosomes, but were localized within multivesicular endosomes and detected in “crude” small EVs fraction. Based on the findings of that paper, authors suggest an exosome-independent amphisome-dependent mechanism of DNA and histone active secretion (Figure 1) [31]. 

Thus, DNA probably finds its way into circulation as a result of cell death and active secretion into the extracellular space. However, it should be noted that the mechanisms responsible for the active secretion of cirDNA, as well as the contributions of different sources of DNA release to the total pool, are still not completely clear.

## 3. CirDNA Forms of Circulation

Several structures involved with cirDNA circulation in biological fluids have been described to date: EVs (small EVs or exosomes, large EVs or microvesicles, apoptotic bodies, etc.) [30], macromolecular complexes (complexes with nucleosomes, other lipids, and proteins, e.g., serum proteins) [32,33], and blood cell surface bound cirDNA [34] (Figure 2). Such association of cirDNA might allow it to be better protected from blood endonucleases, thus allowing for the increase of its half-life in circulation, while the addition of purified deoxyribo-oligonucleotides and DNA amplicons to blood plasma leads to their rapid elimination due to endogenous blood nucleases [35]. 

*Exosomes (small CD9+/CD63+ EVs)*, according to some studies, might take part in the transport of single- and double-helix genome DNA fragments, as well as fragments of mitochondrial DNA. Despite their small size (30–150 nm), exosomes have a large surface area due to their high blood concentration (about 10^7^–10^8^ vesicles/mL). Because the exosomal membrane largely reflects that of the parent cells, similar to cell surface-bound DNA, they are able to transport DNA on their surface. It was found that cirDNA binds to the outer membrane of exosomes by association with DNA-binding proteins. However, the share of exoDNA does not exceed 0.025% and 0.004% in healthy female and breast cancer patients’ plasma DNA, respectively [30].

*Extracellular nanoparticles (ENPs)* of sizes smaller than conventional exosomes might also bind DNA. While the question of DNA abundance in recently discovered superemeres [37] remains open, exomeres from F10, MDA-MB-4175, and AsPC1 cell lines have been shown to carry cirDNA with a fragment length of 100 bp to 10 kb, with a slight enrichment around 2 kb in several cases [38].

*Microvesicles* of a larger size (200–1000 nm) originating from membrane outward budding are also assumed to be able to transfer RNA and DNA from donor cells to recipient cells [39,40]. 

*Apoptotic bodies* are a product of an apoptotic cell death during which the cell splits into large (1 to 5 µm) DNA-containing vesicles, consisting of the cytoplasm and densely packed organelles with or without fragments of the nucleus [41]. The apoptotic bodies’ DNA length is a multiple of 180–200 bp, with a 5′ phosphate and 3′ hydroxyl group [42].

The typical “nucleosomal ladder” characteristic for plasma and serum cirDNA indicates that multimeric complexes of mono- and oligonucleosomes represent the majority of DNA in circulation [10,33]. 

*Nucleosomes* consist of a histone octamer and double-stranded DNA wrapped around this protein complex, stabilized by serum amyloid P [43]. This histone H1 to amyloid P replacement increases the solubility of nucleosomes in plasma. Each nucleosome is linked with the other by a double-stranded DNA linker. The DNA wrapped around the histone octamer is 147 bp in length, while the linker DNA is 20–90 bp long [7]. This association between histones and DNA fragments provides the integrity of the nucleosomal structure, and protects DNA from endonucleases in circulation [44,45]. There is a correlation between the number of cells that have entered apoptosis or necrosis and an increase in the number of nucleosomes in circulation [46]. Moreover, evidently, nucleosomes are released from cells not only under cell death conditions, but also through active secretion [47,48,49,50]. Some authors claim that the majority of DNA circulates in the blood as part of nucleosomes [45]; moreover, the concentration of nucleosomes increases with the progression of pathology [44,45,51]. Currently, several studies have shown that the use of blood-circulating nucleosomes can increase the efficiency of diagnosis and prognosis for non-oncological pathologies such as sepsis, stroke, and autoimmune diseases, as well as malignant neoplasm diagnosis, staging, prediction, and monitoring [45,52]. For example, in patients with colorectal and other types of gastrointestinal cancers, the concentration of nucleosomes correlated with the stage and metastatic status of the disease [33]; a decrease and increase of circulating nucleosomes concentration in patients with remission of the disease and during chemotherapy/radiotherapy, respectively, can be used to monitor the effectiveness of cytotoxic therapy [33].

*Complexes of cell-free DNA with blood proteins.* Since DNA-binding activity is inherent in about 2–3% of serum proteins, extracellular DNA can circulate in complexes with such proteins [5]. These complexes have been described for both major blood proteins such as albumin and immunoglobulins, and for more minor blood proteins such as fibronectin and complement component C1q [53]. Based on the fact that fibronectin, which forms complexes with DNA, is a heparin-binding protein, some authors hypothesize that other proteins of the blood coagulation system that have binding sites for heparin and other polyanions can bind DNA. 

Early studies have shown that the main DNA-binding proteins in blood serum are fractions of high molecular weight proteins (470–760 kDa) and a fraction of proteins with molecular weights of 150–200 kDa [53]. 

Moreover, several blood serum proteins (IgG, albumin, complement component C3, and several apolipoproteins [54]) were shown to be able to bind cirDNA and mediate its opsonization, marking cirDNA for removal from the bloodstream and degradation by nonparenchymal cells [55]. However, in addition to DNA-binding proteins allowing opsonization, there are also proteins that contribute to dysopsonization—increasing the cirDNA half-life in circulation and reducing the rate of its uptake by hepatocytes. Liu et al., in a study of dysopsonin activity of serum DNA-binding proteins, highlight histone H4, PF4, ACTB, ALB, HBA, HBB1, THSD1, and histone-like proteins [55]. The authors show that such proteins are capable of forming complexes with DNA in vivo, protecting it from degradation and nuclease activity without hindering cirDNA biological activity. Thus, serum DNA-binding proteins allowing opsonization/dysopsonization of cirDNA might represent a regulatory mechanism for DNA circulation. 

Lactoferrin and lysozyme were shown to bind DNA in vitro. They were subsequently confirmed to form complexes with a radioactively labeled synthetic oligonucleotide in saliva and tears and, apparently, in the blood. Indeed, lactoferrin concentration in the blood, according to different authors, is 0.05–1.75 µg/mL [56]; it has a high affinity for nucleic acids and binds both ssDNA and dsDNA [57], as well as being able to bind with the cell surface [57] and form complexes with lysozyme [58]. Lysozyme is also capable of binding nucleic acids [7,59,60], and its serum concentration is 5.6–9.2 µg/mL [61].

In a pilot proteomic analysis of affinity chromatography isolated nucleoprotein complexes 111 and 56 proteins were identified in the blood of control donors and breast cancer patients, respectively [32]. The most abundant proteins were HOXC5, GPR22, and IDE, while 40% of identified proteins were characteristic for nucleic acid-/nucleotide-binding and contained several DNA-binding motifs (eight multidomain zinc fingers, two zinc fingers, and five leucine zippers).

*Cell surface bound cirDNA.* The presence of DNA-binding proteins on the outer surface of the plasma membrane indicates cirDNA capability to circulate by binding with blood cells. Such DNA-binding proteins have been described on the leukocyte membrane (20–143 kDa), and the lymphocyte membrane (28 kDa, 59 kDa, 79 kDa) [34]. Moreover, leukocytes were shown not only to carry DNA on their surface, but to also bind it by the ligand–receptor mechanism [34]. Later, the presence of cirDNA, eluted by mild trypsin treatment of the erythrocyte, leukocyte, and platelet surfaces, was confirmed [34,62]. Similar high molecular weight cirDNA fragments were also detected on the cell surface in vitro [27]. Such a large size of fragments can be explained either by a specific secretion mechanism or by the fact that multiple binding to the cell surface more effectively protects long cirDNA fragments from nucleases. Moreover, cirDNA binding with blood cells can occur not only through the nucleic component, but also due to the interaction of protein or lipid components of DNA-containing complexes [63]. For example, the binding of circulating nucleosomes via histones has been described as one of the mechanisms of cirDNA binding to the cell surface [64,65,66]. Membrane receptors are capable of both binding and internalization of cirDNA [67]. While nucleosomes naturally can cross the cell membrane [41,68], some cirDNA-binding proteins, such as albumin, can mediate their internalization via endocytosis [41].

Thus, extracellular DNA can continually circulate in the blood as a component of apoptotic bodies, nucleosomes, and complexes with various proteins. In addition, cirDNA can be associated with the surface of exosomes and blood cells via receptors to nucleic acids and nucleic acid binding proteins. 

## 4. CirDNA Metabolism and Biological Role

Despite the long history of studying circulating nucleic acids and the introduction of methods based on circulating nucleic acids in clinical practice, the role of cirDNA in the organism, both in normal and pathological conditions, remains unclear. 

### 4.1. Immunostimulatory Characteristics of cirDNA

DNA is a macromolecule with immunostimulating properties. This immune response stimulation is based on its double-stranded structure, certain motifs of some sequences, and molecular interactions [1,69]. CirDNA can be perceived by the immune system as a molecular fragment associated with damage, which involves it in the antibacterial and antiviral immune response [36]. Indeed, immune cell interaction with dsDNA leads to the significant activation of genes that regulate the secretion of interferons and other pro-inflammatory molecules. Such stimulation leads to a strong inflammatory response mediated by the secretion of cytokines. CirDNA of nuclear, mitochondrial, and bacterial origin has been shown to similarly stimulate coagulation and platelet activation, but has different effects on inflammation and immune system stimulation [70]. 

CirDNA can activate the immune system both on its own and in combination with other molecules [69,71], and the immunostimulatory effects directly depend on the form of circulation of cirDNA and chromatin (Figure 3) [72]. Histones are cytotoxic for the endothelium and can cause macro- and microvascular thrombosis and renal dysfunction. However, circulating nucleosomes activate different biological pathways upon contact with cells, without the strong cytotoxic effect characteristic of freely circulating histones [36,70,71,72]. Moreover, cirDNA in complexes with histones leads to the induction of anti-DNA antibody production, while the action of blood DNases, on the contrary, inhibits this induction.

There is mounting evidence in the literature that the penetration of cirDNA into the cell and the subsequent triggering of inflammatory biological pathways occurs due to the action of a number of DNA-binding proteins, including histones. Some nucleosome-binding proteins (HMGB1, RAGE) are able to regulate the immunostimulatory effects of both free cirDNA and nucleosomes [73,74,75]. Thus, the form of DNA circulation closely correlates with its biological effects.

### 4.2. Role of cirDNA in Horizontal Gene Transfer 

In addition to the participation of cirDNA in intercellular communication, regulation of inflammation, and the immune system response, various forms of DNA circulation have been described to affect pathophysiological processes associated with the development of malignant neoplasms.

The hypothesis about the transforming ability of cirDNA was first proposed in 1965 [76]. Later, this hypothesis was confirmed in a number of studies and led to the formation of the concept of genometastases: ctDNA is able to end up in healthy cells and lead to malignant transformation [77]. The first work in this field was the transformation of NIH/3T3 mouse fibroblasts in the SW480 culture medium via cirDNA [78]. NIH/3T3 not only went through a malignant transformation after the incubation with SW480 media, but also carried a *KRAS* mutation characteristic for SW480. This effect was also described after the incubation of NIH/3T3 with *KRAS*-positive colorectal cancer patients’ plasma [79]. Moreover, circulating nucleosomal complexes secreted by tumor cells have also been shown to be capable of transferring genetic information to a recipient cell and transforming them into malignant ones. Wang et al., in a 2018 study, demonstrated that, in response to chemotherapy, apoptotic lung cancer cells released HMGB1-containing nucleosomal complexes that mediated tumor invasion and metastasis via TLR4 and TLR9 [80].

Chen et al. [81] suggest that oncogene-containing ctDNA can behave like an oncovirus and transfect normal cells, leading to metastasis (Figure 4A).

This hypothesis expands the concept of “genometastasis”, where the source of oncotransformation is apoptotic bodies’ DNA. Moreover, since there are DNA-binding receptors on the cell surface [34,82,83], the authors suggest tissue-specific metastasis formation In addition, it has been suggested that the cirDNA of normal cells (for example, DNA released into the bloodstream by lymphocytes as a result of antigen stimulation) can transfect tumor cells. In particular, integration of the cytokine-coding region containing cfDNA into a tumor cell genome can lead to the expression of various cytokines, such as interleukin 2, interleukin 12, macrophage colony-stimulating factor, etc. (Figure 4B); cell-free DNA containing an unmutated oncogene (e.g., *ras* gene) or an unmutated oncosuppressor gene (e.g., wild type *p53* gene) can result in knockout via homologous recombination of the corresponding mutant oncogene or suppressor gene within the cancer cell and, consequently, to apoptosis of the tumor cell or even spontaneous remission of cancer (Figure 4B) [84].

The authors [81] explain the phenomenon of the presence of cirNA on the blood cell surface in healthy donors by the ability of cirDNA to bind to receptors on the surface of leukocytes [85], and the decrease in its content during the development of a tumor [81] is explained by the absence of a DNA receptor on the surface of cancer patients blood cells, or loss of DNA-binding properties due to mutation. Apparently, cirDNA is a signaling molecule in the bloodstream, and its binding to a specific receptor on the surface of lymphocytes can lead to cell activation and the emergence of an anti-tumor immune response (Figure 4C). Thus, mutation of the DNA receptor on the surface of lymphocytes can lead to tolerance of the anticancer immune response.

### 4.3. Role of cirDNA in Angiogenesis and Blood Coagulation

In addition to invasion and metastasis, several studies have demonstrated the potential involvement of cirDNA as part of nucleosomal complexes in angiogenesis. Nucleosomes contribute to an increase in the expression of IL-8 (which is involved in the early stages of angiogenesis) by binding to the endothelial cell surface with subsequent activation of the NF-κB/Rel-A pathway [86]. These findings might help explain why hypoxic and hypervascular areas are often found in close proximity in tumor tissues, and may also point to a potential role for nucleosomes in disease progression [45]. Further indirect evidence of the circulating nucleosome participation in angiogenesis is their ability to bind heparin-binding angiogenic factors such as FGF-1, FGF-2, VEGF, and TGFβ-1, stimulating angiogenesis in in vitro and in vivo systems [87].

Recently, cirDNA has been shown to participate in blood coagulation [88]. Evidence that supports this asseveration is that purified genomic DNA increases the activation of proteases that participate in the blood clotting pathway, such as the coagulation factors XII and XI. Moreover, cirDNA from activated neutrophils that are part of the NETs trigger blood clotting that relies on FXII and FXI. Furthermore, it has been observed that histones interact with the A1 domain of the von Willebrand human factor, which can propagate platelet adhesion mediated by GPIbα [88].

### 4.4. Blood cirDNA Clearance

The amount of cirDNA in circulation is determined by the ratio between the release rates and the rates of its internalization, degradation, and elimination [4]. Under the conditions of cancer progression, chronic inflammation, and increased cell death, due to a shift in this ratio towards increased release and insufficient elimination, an increase in the level of cirDNA in the circulation is observed. 

CirDNA half-life in the blood has been estimated by different sources to be from several minutes to up to two hours [36,89,90], and depends on a number of factors, including the form of cirDNA circulation, the type and stage of the disease, the effectiveness of treatment, etc. [1,91]. Degradation and elimination of blood cirDNA are carried out via several mechanisms: degradation by blood plasma endonucleases [92], formation of immunological complexes [93], phagocytosis and lysosomal degradation [13,94,95], metabolism by liver cells [95], and direct elimination of nucleosomal complexes in the kidneys [48,96]. The main role of the blood cirDNA degradation is attributed to circulating enzymes such as DNAse I, FSAP, and factor H [36,97,98]. Moreover, blood proteases have been shown to indirectly affect the levels of nucleoprotein complex cirDNA by hydrolyzing proteins and increasing the availability of nucleic acids for blood nucleases: DNA-protein complexes and protruding DNA pieces are cleaved first, followed by further digestion of better-conserved DNA within nucleosome complexes [92,99,100,101].

Thus, despite the study of the cirDNA phenomenon for more than 70 years, many questions, such as its biological role in circulation and the contribution of various forms of DNA circulation to physiological and pathological processes, remain open.

## 5. CirDNA Perspectives in Liquid Biopsy

Currently, the most topical studies in the field of cirDNA are aimed at evaluating the clinical applicability of cirDNA analysis for diagnosing and monitoring the effectiveness of anticancer therapy [102,103]. Despite the critical importance of non-invasive diagnostic tools in reducing cancer mortality, to date, only one cirDNA-based test has been approved by the FDA in 2016—for the diagnosis of somatic plasma EGFR gene mutations. One of the most serious obstacles to the widespread introduction of such tests in applied oncology is the unsatisfactory signal-to-noise ratio [104]. 

To overcome the hurdles of sensitivity and specificity of such liquid biopsy, knowledge about cirDNA circulation form, its primary structure, mechanisms of secretion, and the contribution of different forms of circulation to the total pool is required. Current limited understanding of the mechanisms by which cirDNA enters the circulation and its further clearance limits the interpretation of existing studies. One approach to increasing the detectability of cirDNA markers is length enrichment. Differences in sizes between cirDNA and ctDNA [105,106,107] suggest that optimization of isolation and processing methods to obtain fragments of a certain size might improve the outcome of the studies.

Another approach to increase the efficiency of liquid biopsy is the use of cirDNA within the multi-marker approach [17,108,109]. For example, the total concentration of cirDNA can serve as a source of information on the status and prognosis of the disease [109,110], and its epigenetic analysis might make it possible to determine hypermethylated regions of tumor genes [63,109], or the cell type that is the source of cirDNA fragments [4,105,111]. Simultaneous analysis of other biomarkers (protein markers from circulating EVs/NPCs, mRNA and miRNA markers from EVs) can provide other levels of information about the prognosis and stage of the disease, response to therapy, and detection of minimal residual disease [112,113], increasing the overall sensitivity and specificity of the approach. Indeed, considerable effort has been devoted to the development of a multi-marker system for cancer diagnosis and monitoring of anticancer therapy, including those using cirDNA. For example, in 2018, a CancerSEEK multi-marker approach was designed based on the cirDNA mutation search and the analysis of proteomic biomarkers (CA-125, CA19-9, CEA, HGF, MPO, OPN, PRL, TIMP-1), which makes it possible to identify eight types of cancer (ovarian, liver, stomach, pancreatic, esophageal, colorectal, lung and breast cancer) with a sensitivity of 55% and a specificity of 99% [114].

Special attention is dedicated to the study of blood-circulating nucleosomes. Several studies have shown that in autoimmune and oncological diseases, use of circulating nucleosome concentration can often increase the efficiency of diagnosis and treatment response monitoring compared to using cirDNA alone [115]. For example, the use of a combination of nucleosome levels and CYFRA 21-1 (the most sensitive non-small lung cancer biomarker) allowed the identification of chemoresistant patients in the cohort of 311 with 29% sensitivity and 100% specificity [116]. Moreover, a recent study showed that, due to differences in the positioning of nucleosomes in different cell types, the profiling of blood plasma cirDNA with an assessment of the pathological status of the cirDNA-secreting cell is possible [111,117,118]. A pilot study shows that this approach allows not only to identify cancers at an early stage and determine the contribution of cirDNA secreted by tumor cells to the total pool, but also to contribute to the understanding of the circulation patterns and structure of nucleosome complexes, both in healthy and oncological conditions, as well as a number of other pathologies, such as heart attack, stroke, autoimmune diseases, etc. [111]. This line of research was continued in the 2019 study, analyzing the activity of transcription factors in the composition of blood-circulating nucleosome complexes from more than 1000 blood samples from healthy donors and colorectal, prostate, and breast cancer patients based on the analysis of cirDNA nucleosome binding patterns. Despite the heterogeneity of cirDNA, this method was shown to be capable of serving as a basis for early non-invasive diagnostic methods, and has been claimed to be more sensitive than multi-marker approaches such as CancerSEEK [119]. 

An alternative approach to improving the results of liquid biopsy may be the development of methods for enriching DNA of tumor origin. In this case, isolating cirDNA using antibodies against tumor-associated proteins in nucleoprotein complexes will make it possible to cut off “noise” from normal DNA, and thus increase the sensitivity of diagnostic systems. 

Another promising approach for noninvasive diagnosis of malignant neoplasms may be the use of tumor-associated proteins of nucleoprotein complexes along with cirDNA analysis. Such a multi-marker approach will improve not only the sensitivity, but also the specificity of non-invasive tests.

Finally, in recent years, there has been increasing evidence that complexes of DNA associated with proteins and lipids are more effective than naked DNA in gene delivery to the nucleus. The knowledge of the composition of proteins that interact with cirDNA will provide a better understanding of the homeostasis of circulating nucleic acids and the different interactions with several target cells that may be useful in developing gene-target therapy approaches.

## Figures and Tables

**Figure 1 ijms-23-07224-f001:**
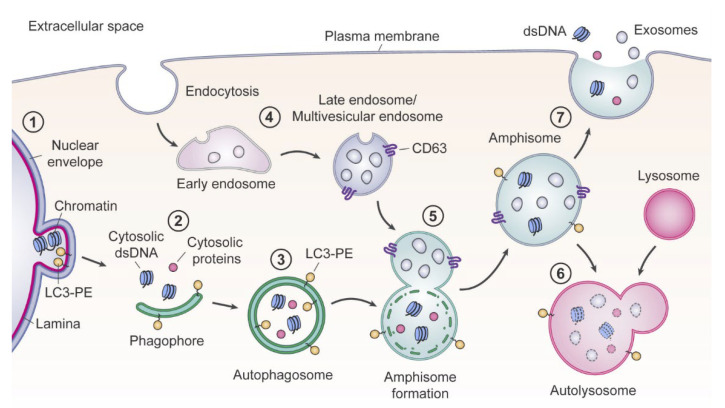
Jeppesen et al. model of amphisome-dependent, exosome-independent secretion. For autophagy, cytosolic LC3 is lipidated by conjugation with phosphatidylethanolamine to form LC3-PE. (1) Nuclear membranes can bleb in a process dependent on LC3B and the nuclear lamina protein Lamin B1, causing the appearance of cytoplasmic chromatin fragments. (2) During autophagy, cytoplasmic components are sequestered as a phagophore begins to engulf material. (3) Continued expansion of autophagic membranes requires LC3-PE and results in formation of the double-membrane autophagosome. (4) As early endosomes develop to late endosomes, the pH decreases and continued invagination of limiting membranes generates intraluminal vesicles (ILVs). A fully developed CD63-positive multivesicular endosome (MVE) contains numerous ILVs. (5) Fusion of the autophagosome with a MVE causes degradation of the inner autophagosome membrane, generating an amphisome, a single-membrane hybrid compartment. (6) The amphisome fuses with a lysosomal compartment to form the autolysosome followed by degradation of cargo, or alternatively, (7) the amphisome fuses with the plasma membrane causing extracellular release of dsDNA and histones, and separately, the ILVs as exosomes. Reprinted with permission from Ref. [31]. Copyright 2019, with permission from Elsevier.

**Figure 2 ijms-23-07224-f002:**
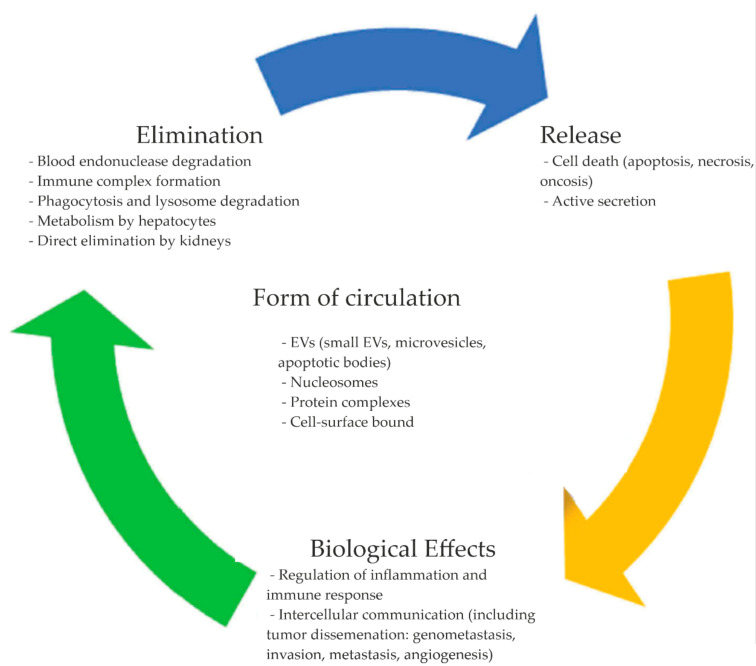
Current views on cirDNA origins and fate. Reprinted with permission from Ref. [36] Copyright 2019 The Author(s). Published with license by Taylor & Francis Group, LLC.

**Figure 3 ijms-23-07224-f003:**
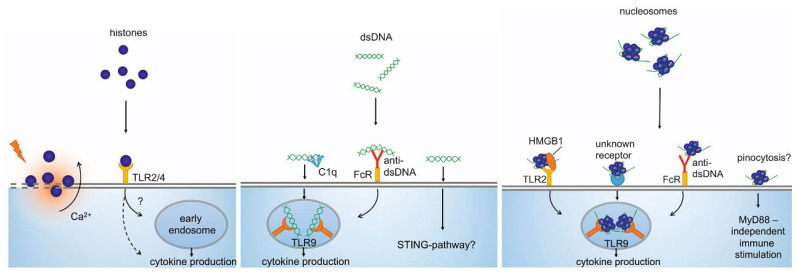
Differences in cirDNA, nucleosome and histone internalization, and their following biological effects. Reprinted with permission from Ref. [72]. Copyright 2016 The Author(s) Published with license by Nature Publishing Group.

**Figure 4 ijms-23-07224-f004:**
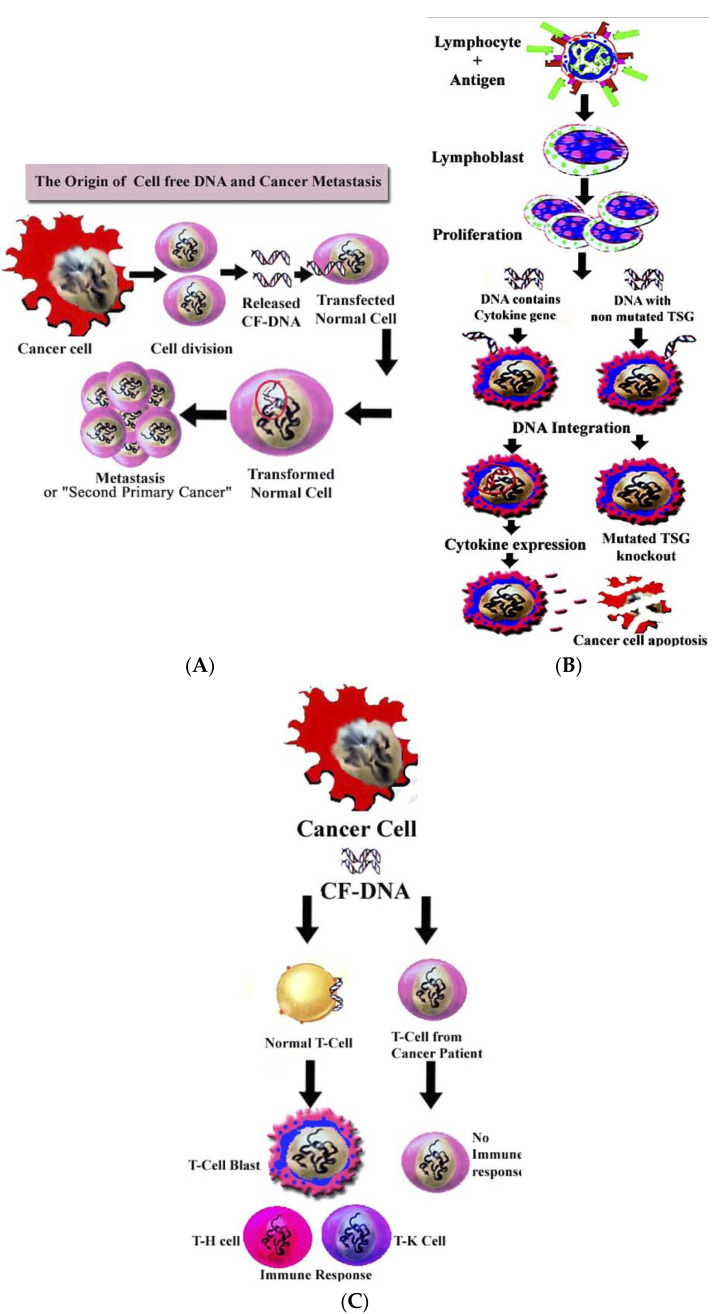
(**A**) The origin of cf-DNA and cancer metastasis. Cancer cells release DNA when they divide. The released DNA (cf-DNA) circulates in the body fluids and has the ability to transfect and transform adjacent or remote normal cells. The transformed cells may keep growing, and so cancer metastases or second primary cancer develops. (**B**) Hypotheses of two possible outcomes if the lymphocyte DNA transfects cancer cells. If this DNA contains cytokine’s functional coding region, the transfected cancer cells may become cytokine releasing cells and, in some cases, may work like an intrinsic cancer vaccine releasing therapeutic cytokine (left). If this DNA contains a coding region for a nonmutated oncogene or tumor suppressor gene, a homologous DNA recombination may cause knock out of a correspondent mutated oncogene or tumor suppressor gene that may lead to cancer cell apoptosis, and thus a dramatic spontaneous remission of the cancer may ensue (right). (**C**) Hypotheses of two possible effects of cancer cf-DNA on lymphocytes. If the lymphocyte contains the functional DNA receptor, cancer cf-DNA binding to the receptor may activate the lymphocyte, thus forming lymphoblasts that proliferate and differentiate into immune responding cells (e.g., helper-T cells or killer-T cells) to enhance immune function (left). If the lymphocyte contains no functional DNA receptor, the cf-DNA has no effect on the lymphocyte without any change in immune response (right) [81]. Copyright 2005, with permission from Elsevier.

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
