# Peer review of "The Influence of Proteins on Fate and Biological Role of Circulating DNA"

_ijms, 2022, doi:10.3390/ijms23137224_

Round 1

Reviewer 1 Report

The review from Tutanov and Tumkovich deals with a great interest topic, often neglected, about the potential role of proteins associated with circulating DNA.  

However, I have some remarks and  questions :  

  1. I think the acronym «â€¯cirDNA » is ambiguous because it can refer to circular DNA, maybe the acronym "cfDNA" (Cell-free DNA) usually  used should be a better choice as you defined it in figure 4. Alternatively, please define these terms. 

  1. The review starts logically with the origin of circulating DNA.  After reading this paragraph, my feeling is that we are not sure which is the strongest hypothesis with respect to the bibliography. The paragraph would gain in clarity by structuring the hypotheses according to the origin of circulating DNA: i) intracellular: cell death (apoptosis vs necrosis) or secreted by living cells (exosome/amphisome), ii) fragmentation in the ECM , iii) fragmentation in blood.  

  1. a) What is the role of the different nucleases in these compartments?  

  1. b) Role of histone H3 on fragment size is unclear for me. Could you comment ?  

  1. c) Does the increase of autophagy increase the concentration of circulating DNA? 

I agree with the authors that the presence of DNA in exosomes is a tricky topic. It is indeed difficult to understand how a cell could actively secrete DNA! The authors explain that the DNA could be transported on the membrane of the exosomes via proteins already identified. This is probably a more likely explanation.  Is the function of these proteins known and would it allow a targeted transport of DNA? 

  1.  

Generally speaking, chapters take too long to read in one block (especially Chapter 4). Perhaps the reading would be more fluid by adding sub-chapters and relying on the figures. 

  1. The review ends wisely with potential applications in liquid biopsy. The main question is to what extent proteins associated with DNA shoul be taken into consideration for the analysis of circulating DNA ? Authors have partially answer to this question : modification of size, kinetics...maybe a few sentences on how to use these informations in future could be of interest ie which analaysis of proteins associated with circulating DNA could have a medical interest: position of nucleosomes, nature of histones, other protein complexes ? 

In conclusion the authors have done a lot of bibliography work which is generally interesting.   I think however it is necessary to better structure the ideas in a more logical way in order to faciltate the reading

Author Response

To Reviewer 1: 

Comments and Suggestions for Authors

The review from Tutanov and Tumkovich deals with a great interest topic, often neglected, about the potential role of proteins associated with circulating DNA.  

However, I have some remarks and  questions :  

I think the acronym «â€¯cirDNA » is ambiguous because it can refer to circular DNA, maybe the acronym "cfDNA" (Cell-free DNA) usually  used should be a better choice as you defined it in figure 4. Alternatively, please define these terms. 

Response: We fully agree with the reviewer's opinion that acronym “cirDNA” (circulating DNA) is similar to circular DNA. However, in that case, the acronym “circDNA” is used.

The acronym cirDNA is widely used in the field, as can be found in such basic overviews as

* Origins, structures, and functions of circulating DNA in oncology. Thierry AR, El Messaoudi S, Gahan PB, Anker P, Stroun M. Cancer Metastasis Rev. 2016;35(3):347-76. doi: 10.1007/s10555-016-9629-x.

** A historical and evolutionary perspective on the biological significance of circulating DNA and extracellular vesicles. Aucamp J, Bronkhorst AJ, Badenhorst CP, Pretorius PJ. Cell Mol Life Sci. 2016;73(23):4355-4381. doi: 10.1007/s00018-016-2370-3

We agree that the use of terms cell-free DNA or extracellular DNA also occurs in the scientific literature. However, since in this review we consider not only cell-free DNA, but also cell-surface-bound DNA, we wanted to use the acronym “cirDNA” to combine both cfDNA and csbDNA. We thought that the use of the acronym “cfDNA” in relation to cell-surface-bound DNA will confuse readers.

We have not made any changes to Figure 4 because we have quoted it.

We added to page 1 definition of term cirDNA.

The review starts logically with the origin of circulating DNA.  After reading this paragraph, my feeling is that we are not sure which is the strongest hypothesis with respect to the bibliography. The paragraph would gain in clarity by structuring the hypotheses according to the origin of circulating DNA: i) intracellular: cell death (apoptosis vs necrosis) or secreted by living cells (exosome/amphisome), ii) fragmentation in the ECM , iii) fragmentation in blood.  

  1. a) What is the role of the different nucleases in these compartments

Response: We thank the reviewer-1 for the recommendation on structuring the section “Origin of cirDNA”. Since additional DNA degradation occurs after entering the bloodstream (whether it’s from dying or living actively secreting cells), we have divided this section into the following sub-chapters (2.1. Cell Death (Apoptosis and Necrosis); 2.2. Appearance of Blood cirDNA During Secretion by Normal and Tumor Cells), somewhat expanding its contents. The section “fragmentation in the ECM” is not added due to the lack of sufficient information to form a separate section.

We have not added the “Fragmentation in blood” subsection to the "Origins of cirDNA" section. We think that fragmentation of secreted DNA (in forms of nuclueosomes and complexes with DNA-binding proteins) in the blood mainly occurs due to nucleases (DNase I, DNase II, phosphodiesterase I, etc.). Therefore, fragmentation by nucleases is mentioned in section 2 and discussed in more detail in section 3.

  1. b) Role of histone H3 on fragment size is unclear for me. Could you comment ? 

Response: Packaging of DNA into a histone core affects the nature of DNA fragmentation. In particular, it has been shown that the linker part of the nucleosome is more accessible for hydrolysis. In the cited work (Jeppesen, D.K., Cell 2019, 177, 428-445) the authors identified the composition of non-vesicular particles secreted by cells. It has been shown for the DNA to be associated with histones. Then, using immunoblotting, they verified MS data. Since the authors used antibodies against H2a and H3, we cite only verified data in our review.

  1. c) Does the increase of autophagy increase the concentration of circulating DNA?

Response: We thank the reviewer for reminding us about the phenomenon of autophagy and the possible contribution of this process to increasing concentration of cirDNA. At the same time, as in the case of apoptosis and necrosis, this cellular debris must also be phagocyted by macrophages. Nevertheless, it can be assumed that some part of the cellular debris can avoid this fate and remains in the blood, increasing the concentration of cirDNA. We added this type of cell death to new subsection “Cell Death Origin of cirDNA”

I agree with the authors that the presence of DNA in exosomes is a tricky topic. It is indeed difficult to understand how a cell could actively secrete DNA! The authors explain that the DNA could be transported on the membrane of the exosomes via proteins already identified. This is probably a more likely explanation.  Is the function of these proteins known and would it allow a targeted transport of DNA? 

Response: Identification of proteins involved in DNA transport both as part of exosomes and as part of nucleoprotein complexes has been started in our laboratory recently. The article on the identification of DNA-binding proteins is a pilot (Tutanov, et al. Blood Plasma Exosomes Contain Circulating DNA in Their Crown. Diagnostics 2022, 12, 854). We hope to get answers to this and other questions before the end of this year.

Generally speaking, chapters take too long to read in one block (especially Chapter 4). Perhaps the reading would be more fluid by adding sub-chapters and relying on the figures. 

Response: In accordance with the recommendation, we have divided sections « Origin of cirDNA” and « CirDNA metabolism and biological role” into subsections. In section « CirDNA forms of circulation”, there is already a semantic division according to the forms of cirDNA circulation (italicized).

The review ends wisely with potential applications in liquid biopsy. The main question is to what extent proteins associated with DNA shoul be taken into consideration for the analysis of circulating DNA ? Authors have partially answer to this question : modification of size, kinetics...maybe a few sentences on how to use these informations in future could be of interest ie which analaysis of proteins associated with circulating DNA could have a medical interest: position of nucleosomes, nature of histones, other protein complexes ? 

Response: In accordance with the recommendations of the reviewer-1, we have expanded the last section. We suggest that information about tumor-associated proteins in circulating complexes will significantly improve liquid biopsy. In particular, the enrichment of tumor DNA by isolating cirDNA using antibodies against tumor-associated proteins in nucleoprotein complexes will make it possible to cut off "noise" from normal DNA and thus increase the sensitivity of diagnostic systems. In addition, the use of a multimarker approach for non-invasive diagnostics based on the analysis of tumor-associated proteins of nucleoprotein complexes, along with the analysis of cirDNA, will improve not only the sensitivity, but also the specificity of non-invasive tests.

Finally, we hope that knowledge of the composition of proteins interacting with cirDNA will allow a better understanding of the homeostasis of circulating nucleic acids and their interaction with several target cells, which may be useful for developing gene-target therapy approaches.

In conclusion the authors have done a lot of bibliography work which is generally interesting.   I think however it is necessary to better structure the ideas in a more logical way in order to faciltate the reading. 

Response: We thank you for the valuable and useful comments aiming to improve the manuscript.

Sincerely yours,

Svetlana Tamkovich

Reviewer 2 Report

As far as I understand the authors (two ladies team) they attempted to damp partly a common enthusiasm associated with application of circulating DNA as clinical biomarker. The article is not written against usage of cir DNA for clinical purposes. Nevertheless, the authors are trying to expose several uncertainties connected with an origin of cirDNA in body fluids. Further an attention is paid on unstable concentration of cirDNA associated with interaction with proteins including proteases. Thus the results derived from measurements of cirDNA levels should be carefully estimated.

The article is well written concerning both the matter and language. Very digestive presentation seems to be derived from a considerable experience of the authors in cirDNA studies.

Author Response

To Reviewer 2: 

English language and style

( ) Extensive editing of English language and style required
(x) Moderate English changes required
( ) English language and style are fine/minor spell check required
( ) I don't feel qualified to judge about the English language and style

Response: In the revised version of the manuscript, the English has been verified by the translator of the Institute.

Comments and Suggestions for Authors

As far as I understand the authors (two ladies team) they attempted to damp partly a common enthusiasm associated with application of circulating DNA as clinical biomarker. The article is not written against usage of cir DNA for clinical purposes. Nevertheless, the authors are trying to expose several uncertainties connected with an origin of cirDNA in body fluids. Further an attention is paid on unstable concentration of cirDNA associated with interaction with proteins including proteases. Thus the results derived from measurements of cirDNA levels should be carefully estimated.

The article is well written concerning both the matter and language. Very digestive presentation seems to be derived from a considerable experience of the authors in cirDNA studies.

Response: We sincerely thank the reviewer-2 for the high assessment of the work done. Indeed, I have been working in the field of cirDNA research since 2000, and Oleg in the field of proteomics since 2013.

Sincerely yours,

Svetlana Tamkovich

Reviewer 3 Report

This manuscript is comprehensive review of circulating DNA, focusing on the roles of protein-DNA complex. I would strongly suggest the acceptance of this work in International Journal of Molecular Sciences.

Minor comments,

Figures used are the copies from the original papers. I don’t know the publication policy of IJMS, but have the authors obtained the permissions from publishers?

Some readers may not know the differences of definition between “exosome” and “small EV”. It would be helpful to mention the relationships between them.

In Introduction, the authors described “First demonstrations of DNA in the blood of healthy individuals date back to 1948”, and in Section 3, they stated “The phenomenon of small amounts of DNA circulating in the blood of healthy hu-103 mans and animals was reported back in the 1960s”, which may confuse readers. Some modification in the description is required.

Author Response

To Reviewer 3: 

Comments and Suggestions for Authors

This manuscript is comprehensive review of circulating DNA, focusing on the roles of protein-DNA complex. I would strongly suggest the acceptance of this work in International Journal of Molecular Sciences.

Response: We sincerely thank the reviewer-3 for the high assessment of the work done.

Minor comments,

Figures used are the copies from the original papers. I don’t know the publication policy of IJMS, but have the authors obtained the permissions from publishers?

Response: We have already obtained permissions to copy the figures from the respective publishers.

Some readers may not know the differences of definition between “exosome” and “small EV”. It would be helpful to mention the relationships between them.

Response: In accordance with the recommendation, we have added an explanation of the term "exosome" on page 2:

Currently, in the fraction of small EVs is distinguished exosomes - vesicles with a size of 30-150 nm, carrying such tetraspanins as CD 9, CD63 and CD81 on their membrane.

In Introduction, the authors described “First demonstrations of DNA in the blood of healthy individuals date back to 1948”, and in Section 3, they stated “The phenomenon of small amounts of DNA circulating in the blood of healthy hu-103 mans and animals was reported back in the 1960s”, which may confuse readers. Some modification in the description is required.

 Response: We fully agree with the reviewer's opinion about the ambiguity of the wording in the introduction and section 3. Indeed, the first demonstration of cirDNA and cirRNA in the blood of healthy people and sick patients (ischemia, diabetes, etc.) was carried out by Mendel and Met in 1948. Unfortunately, this phenomenon was ignored and re-attracted the attention of researchers in 1966 only. We have removed the first sentence of section 3 as having little to do with the section “CirDNA forms of circulation”

We thank you for the valuable and useful comments aiming to improve the manuscript.

Sincerely yours,

Svetlana Tamkovich

Round 2

Reviewer 1 Report

I read with interest author's answers to my questions. This is a really interesting topic and there is no doubt about the interest of  this detailed review for IJMS readers. I appreciate the changes that have been made to the document which now seems ready for publication.